# Spurious precision in meta-analysis of observational research

Zuzana Irsova [1,2] ✉, Pedro R. D. Bom[3], Tomas Havranek [1,2,4] &
Heiko Rachinger[5]

Meta-analysis assigns more weight to studies with smaller standard errors to maximize the precision of the overall estimate. In observational settings, however, standard errors are shaped by methodological decisions. These decisions can interact with publication bias and *p*-hacking, potentially leading to spuriously precise results reported by primary studies. Here we show that such spurious precision undermines standard meta-analytic techniques, including inverse-variance weighting and bias corrections based on the funnel plot. Through simulations and large-scale empirical applications, we find that selection models do not resolve the issue. In some cases, a simple unweighted mean of reported estimates outperforms widely used correction methods. We introduce MAIVE (Meta-Analysis Instrumental Variable Estimator), an approach that reduces bias by using sample size as an instrument for reported precision. MAIVE offers a simple and robust solution for improving the reliability of meta-analyses in the presence of spurious precision.

Inverse-variance weighting reigns in meta-analysis[1]. More precise studies, or rather those seemingly more precise based on lower reported standard errors, get a greater weight explicitly or implicitly. The weight is explicit in traditional summaries, such as the fixed-effect model (assuming a common effect) and the random-effects model (allowing for heterogeneity)[2,3]. They are weighted averages, with the weight diluted in random effects by a heterogeneity term.

The weight is also explicit in publication bias correction models based on the funnel plot[4–9]. In funnel-based models, reported precision is especially crucial because the weighted average gets reinforced by assigning more importance to supposedly less biased (nominally more precise) studies. The weight is implicit in selection models estimated with maximum likelihood[10–15], which, in the absence of publication bias, often reduce to the random-effects model.

The tacit assumption behind these techniques is that the reported precision accurately reflects the study's uncertainty. In other words: the standard error is given to the researcher by her data and methods. It is fixed and cannot be manipulated, consciously or unconsciously. The assumption is perhaps plausible in experimental research, for which most meta-analysis methods were developed. But

in observational research, which comprises the vast majority of the scientific literature[16,17] and on which thousands of meta-analyses are conducted each year, the derivation of the standard error is often a key part of the empirical exercise.

Consider, for example, a regression analysis with longitudinal data: explaining the educational outcomes of children taught by different teachers and observed over several years. Individual observations are not independent, and standard errors in the regression need to be clustered[18]. But how? At the level of teachers, classes, schools, children, or years? Should one use double clustering[19] or perhaps wild bootstrap[20]? It is complicated[21,22], and with a different clustering/bootstrapping choice the researcher will report different precision.

Spurious precision can arise in many contexts other than longitudinal data analysis. Ordinary least squares, the workhorse of observational research, assumes homoskedasticity of errors. The assumption is often violated, and researchers should use robust standard errors[23], typically larger than unadjusted ones. If researchers ignore heteroskedasticity, they report spuriously precise estimates. Moreover, heteroskedasticity- and cluster-robust standard errors are often downward biased in small- and medium-sized samples[24,25], and

[1]Institute of Economic Studies, Faculty of Social Sciences, Charles University, Prague, Czech Republic. [2]Meta-Research Innovation Center at Stanford, Stanford University, Stanford, CA, USA. [3]University of Deusto, Bilbao, Spain. [4]Centre for Economic Policy Research, London, UK. [5]University of the Balearic Islands, Palma, Spain. ✉e-mail: zuzana.irsova@fsv.cuni.cz

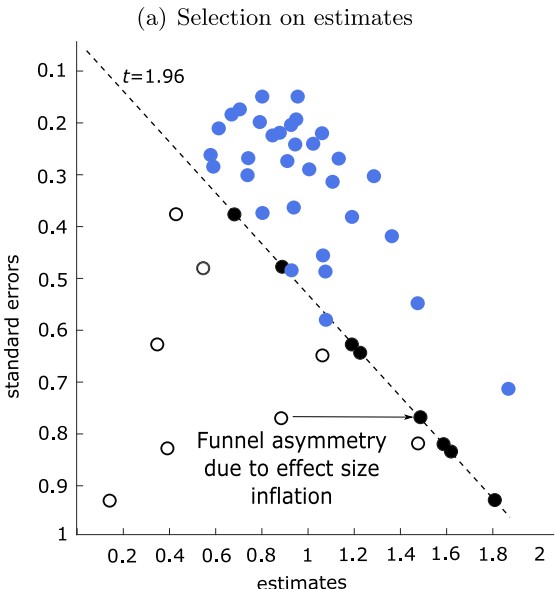

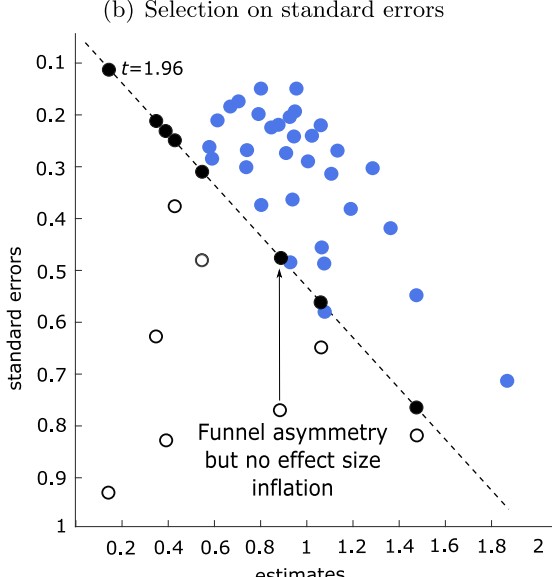

**Fig. 1 | Two flavors of selection and repercussions for meta-analysis.** Notes: Blue-filled circles (lighter in grayscale) denote estimates statistically significant at the 5% level; these are reported. Hollow circles denote insignificant estimates, which are not reported in their original form but *p*-hacked to yield statistical significance (black-filled circles). In panel (**a**) the resulting mean of reported estimates is biased upwards, and inverse-variance weighting helps mitigate the bias. In panel (**b**) the resulting mean is unbiased, and inverse-variance weighting introduces a bias. A realistic scenario of *p*-hacking combines both types of selection, so the *p*-hacked estimates move not strictly east or north (as in the figure) but northeast.

the same problem applies to classical standard errors in instrumental-variable and difference-in-differences models, even with large samples[26,27]. Reported standard errors may also be underestimated in experiments due to violations of randomization assumptions, finite sample issues, or model misspecifications[28]. When a study with spurious precision enters meta-analysis, it wields exaggerated impact due to inverse-variance weighting.

In this paper we use simulations and large-scale applications to show that spurious precision can create serious problems for meta-analysis. We offer a solution that removes most of the resulting bias by adjusting the reported precision to more closely reflect the sample size used in the primary study. The proposed approach uses two-stage least squares and we call it Meta-Analysis Instrumental Variable Estimator (MAIVE).

## Results

### Mechanisms of spurious precision

Spurious precision could potentially emerge due to manipulation of data or reported standard errors. For economics journals, quasi-experimental evidence shows that the introduction of obligatory data sharing reduced the reported *t*-statistics[29]. These results imply that, before data sharing was mandated, some authors may have manipulated data or results to some extent—perhaps by removing "outliers" and thereby decreasing reported uncertainty. But a more realistic source of spurious precision is *p*-hacking, in which the researcher can sometimes adjust the entire model (e.g., by changing control variables in a regression setting) to produce statistically significant results. After adjusting the model, both the effect size and standard error change, and both changes can jointly contribute to statistical significance. We examine, by employing Monte Carlo simulations and empirical applications, the mechanisms and importance of spurious precision.

Figure 1 gives intuition on a simulation scenario inspired by potential clustering/heteroskedasticity/manipulation context discussed above. For brevity we call it a stylized scenario. Here researchers crave statistical significance and to this end change reported effect sizes or standard errors, but not both at the same time. The scenario is simplistic and meant to illustrate the concept of spurious precision, not capture the real behavior of researchers. We start with it because it allows for a transparent separation of selection on estimates (conventional in the literature) and selection on standard errors (our focus). The separation is less clean in the more realistic *p*-hacking scenario but can be mapped back to the stylized scenario. Note that both differ from the standard scenario of publication bias, where different studies have different probabilities of publication but are not manipulated or *p*-hacked.

The mechanism of the left-hand panel of Fig. 1 is analogous to the Lombard effect in psychoacoustics[30]: speakers increase their vocal effort in response to noise. Here researchers increase their selection effort in response to noise in data or methods, noise that produces insignificance. When selection works on effect sizes, the results are consistent with funnel-based models of publication bias: funnel asymmetry arises, the most precise estimates are close to the true effect, and inverse-variance weighting helps mitigate the bias.

By contrast, the mechanism of the right-hand panel of Fig. 1 is analogous to Taylor's law in ecology[31]: variance can decrease in response to a smaller mean. When researchers achieve significance by lowering the standard error to compensate for a small estimated effect size, we again observe funnel asymmetry. But this time no bias arises in the reported effect sizes. The simple unweighted mean of reported estimates is unbiased, and it is inverse-variance weighting that creates a bias. The bias increases when we use a correction based on the funnel plot: that is, when we estimate the effect size of a hypothetical infinitely precise study.

In practice, as noted, selection on estimates and standard errors likely arises simultaneously. We generate this quality in simulations by allowing researchers to replace control variables in a regression context. Control variables are correlated with the main regressor of interest, and their replacement affects both the estimated effect in question and the corresponding precision. Then *p*-hacked estimates move not strictly east or north, as in the figure, but northeast. Spuriously large estimates can then be also spuriously precise. Our simulations and applications suggest that an upward bias arises in conventional meta-analysis estimators, including those that try to correct for publication bias.

## Formalization and conceptual framework

Suppose that the object of meta-analysis is $\alpha_1$, the slope coefficient of a regression of an outcome variable $Y$ on a predictor $X_1$, after controlling for covariates $X_2, …, X_p$:

$$Y_j = \alpha_0 + \alpha_1 X_{1j} + \alpha_2 X_{2j} + … + \alpha_p X_{pj} + u_j, \tag{1}$$

where $j$ denotes individual observations and $u$ is an error term with zero mean conditional on $X_1, …, X_p$. For example, if the interest lies in the effect of teacher experience on the academic performance of pupils, $Y$ would measure the pupil's test score, $X_1$ would measure the teacher's experience, and it would be necessary to control for confounding factors such as socioeconomic background (children from families with higher socioeconomic status may simultaneously show better test scores and attend schools where teachers have more experience).

Assume that primary studies estimate a form of (1), always featuring $Y$ and $X_1$, but potentially omitting some or all covariates $X_2, …, X_p$. The estimates of $\alpha_1$ and the corresponding standard errors reported in these studies provide the data for meta-analysis. The precision of an estimate is the inverse of its reported standard error. For simplicity, we assume that all primary studies define $Y$ and $X_1$ on the same scale, so that no rescaling is necessary at the meta-analysis level. Note that while we work with comparable regression coefficients to avoid the mechanical correlation between estimates and standard errors that arises with standardized effect sizes, MAIVE can be applied in settings involving standardized effect sizes. In fact, this mechanical correlation is another reason why MAIVE is particularly useful in such cases.

The ordinary least squares (OLS) estimator provides an unbiased and consistent estimate of $\alpha_1$ in model (1). The corresponding standard error is given by:

$$\text{SE}(\hat{\alpha}_1) = \sqrt{\frac{s^2}{(N-1)\,\text{Var}(X_1)(1 - R_1^2)}}, \tag{2}$$

which depends on the estimated variance of $u$ ($s^2$), the sample size ($N$), the variance of $X_1$, and the correlation between $X_1$ and the other regressors (measured by $R_1^2$).

Equation (2) hinges on the absence of heteroskedasticity and autocorrelation. If the error term is heteroskedastic or exhibits a clustered structure of correlation, (2) usually underestimates the true standard error of $\alpha_1$. This misspecification of the statistical properties of the error term is one potential cause of spurious precision. Another potential cause is the misspecification of the functional form of the regression, a mechanism we focus on in simulations. Suppose, for

example, that one or more of the control variables $X_2, …, X_p$ are correlated with $X_1$ but are excluded from (1), causing omitted-variable bias. Then $R_1^2$ decreases and $s^2$ increases, affecting both the numerator and denominator of (2). We focus on the more intuitive case in which the former effect dominates, so the resulting standard error is smaller than the one obtained from the well-specified model (1).

Reported precision is defined as spurious if it exceeds that in a correctly specified model with proper functional form and error term properties. Correct precision, by contrast, requires that the model yields an unbiased and consistent estimate of $\alpha_1$ and that the computation of the standard error accounts for clustering or small-sample issues when present. Spurious precision thus defined is relevant at the meta-analysis level because it undermines the effectiveness of conventional methods, which use the standard error as inverse weight and also as a regressor in Egger-type regression, to identify the true mean effect beyond publication bias.

Spurious precision by itself does not bias meta-analysis if it is unrelated to the size of the estimated effect; in that case, it only reduces efficiency and distorts inference. But it becomes a more serious problem when some authors crave statistical significance and adjust the computation of precision in response to the estimated effect size—even if the corresponding estimate reported in the primary study remains unbiased (as we will see in the stylized simulation scenario). Spurious precision is particularly detrimental to meta-analysis when it co-occurs with a biased estimate (as we will see in the $p$-hacking simulation scenario). Note that the standard error of a biased estimate of $\alpha_1$ from a misspecified model may be regarded as correct for that particular model. However, at the meta-analysis level, cross-study comparison of precision matters, and in this case a biased estimate would be paired with higher precision—and thus greater weight—than if it were unbiased.

## Conventional meta-analysis models

We focus on minimizing bias in meta-analysis, though in the Supplementary Information (S1, S6) we also report and discuss results for MSE and coverage rates. Does any meta-analysis technique work well in the case of panel B of Fig. 1, or at least with a small amount of spurious precision? We examine seven current estimators: simple unweighted mean, fixed-effect model (weighted least squares, FE/WLS)[32], precision-effect test and precision-effect estimate with standard errors (PET-PEESE)[7], endogenous kink (EK)[9], weighted average of adequately powered estimates (WAAP)[8], the model by Andrews and Kasy[14], and $p$-uniform*[15]. Note that the unweighted mean can be seen as a limiting case of a random-effects model, where between-study variance is assumed to be infinite and all estimates are weighted equally.

The first two are summary statistics, the next three are corrections based on the funnel plot, and the last two are selection models. The three funnel-based techniques are commonly used in observational research[33–38]. The two selection models, based on random-effects specifications, are also used often[39–45] and represent the latest incarnations of models in the tradition of Hedges[10–13] and their simplifications[46–51].

The importance of reported precision for these estimators is summarized in Table 1. In most of them precision has two roles: weight and identification. Identification, the unique determination of the mean true effect in meta-analysis based on observed primary studies, can be achieved through Egger-type regression (where the standard error or a function thereof is included as a regressor), selection model, or a combination of both—such as the EK model.

None of these estimators perform well with even modest spurious precision working through selection on standard errors. The reader might expect selection models to beat funnel-based models, because of the latter's heavier reliance on precision. This is not always the case, and even selection models are sometimes defeated by the unweighted mean when selection on standard errors is modest (about 1:3

**Table 1 | The role of the standard error in 7 benchmark estimators**

| Estimator | Weight | Regressor | Identification |
|---|---|---|---|
| Simple average | | | |
| FE/WLS | ✓ | | |
| PET-PEESE | ✓ | ✓ | |
| EK | ✓ | ✓ | ✓ |
| WAAP | ✓ | | ✓ |
| Andrews & Kasy | ✓ | | ✓ |
| $p$-uniform* | ✓ | | ✓ |

Notes: Simple average = unweighted mean. FE/WLS = fixed effect/weighted least squares: mean weighted by inverse variance[32]. PET-PEESE = precision-effect test and precision-effect estimate with standard errors: selection is a quadratic function of SE when true effect $\neq 0$[7]. EK = endogenous kink: selection is a linear function of SE for imprecise estimates, no selection for precise ones[9]. WAAP = weighted average of adequately powered estimates: only estimates with at least 80% power included[8]. Andrews & Kasy = a selection model in the tradition of Hedges[14]. $p$-uniform* = a simplified selection model based on the principle that $p$-values should be uniformly distributed at the true effect size[15].

**Table 2 | Estimators and their MAIVE variants considered in simulations**

| Estimator | Variants |
| --- | --- |
| Simple average | (1) Unadjusted |
| FE/WLS | (1) Unadjusted |
|  | (2) Adjusted weights |
| PET-PEESE | (1) Unadjusted |
|  | (2) Adjusted weights |
|  | (3) Instrumented SEs |
|  | (4) Adjusted weights and instrumented SEs |
|  | (5) Instrumented SEs and no weights (MAIVE baseline) |
| EK | (1) Unadjusted |
|  | (2) Adjusted weights |
|  | (3) Instrumented SEs |
|  | (4) Adjusted weights and instrumented SEs |
|  | (5) Instrumented SEs and no weights |
| WAAP | (1) Unadjusted |
|  | (2) Adjusted weights and SEs |
| Andrews & Kasy | (1) Unadjusted |
|  | (2) Adjusted SEs |
| $p$-uniform* | (1) Unadjusted |
|  | (2) Adjusted SEs |

Notes: See notes to Table 1.

compared to selection on estimates). The bias in meta-analysis due to spurious precision can be more important than the classical publication bias.

## Meta-analysis instrumental variable estimator

We propose a straightforward adjustment of current meta-analysis techniques, the meta-analysis instrumental variable estimator (MAIVE), which corrects most of the bias and restores valid coverage rates. MAIVE replaces, in all meta-analytic contexts, the reported variance with the portion that can be explained by the inverse of the total sample size used in the primary study, regardless of the model specification in the original studies (e.g., whether they use multilevel or clustered designs). MAIVE relies on the following regression:

$$SE(\hat{\alpha})_i^2 = \psi_0 + \psi_1(1/N_i) + \nu_i, \tag{3}$$

where $\hat{\alpha}$ is the effect size reported in a primary study, $SE$ is its standard error, $\psi_0$ is a constant, $N_i$ is the sample size of the primary study, and $\nu_i$ is an error term that soaks up, among other things, the spurious elements of the reported standard error related to $p$-hacking. We explain the intuition behind MAIVE by starting with a version of the Egger regression[4]:

$$\hat{\alpha}_i = \alpha_0 + \beta SE(\hat{\alpha})_i^2 + \nu_i. \tag{4}$$

This is the PEESE model, but for simplicity without additional inverse-variance weights—since the model searches for the effect conditional on maximum precision ($\alpha_0$), it already features an implicit, built-in weight.

In panel A of Fig. 1, the quadratic regression (PEESE, Eq. (4)) would fit the data quite well[7], and estimated $\alpha_0$ would lie close to the mean underlying effect. In panel B, however, PEESE fails to recover the underlying coefficients. PEESE fails because it assumes a causal effect of the standard error on the estimate: a good description of panel A (Lombard effect), but not panel B (Taylor's law). In panel B, the standard error sometimes depends on the estimated effect size and is thus correlated with the error term, $\nu_i$. In other words, spurious precision breaks the exogeneity assumption in (4) that is necessary for ordinary least squares to correctly identify regression parameters. The resulting estimates of $\alpha_0$ (true effect) and $\beta$ (intensity of selection) are biased.

The problem is the correlation between $SE$ and $\nu_i$, which can potentially arise for four reasons: First, selective reporting based on standard errors, which we simulate. Second, measurement error in $SE$[52]. Third, mechanical relation between estimates and standard errors in standardized effect sizes[37,53,54]. We do not consider these two sources of correlation in simulations. Fourth, unobserved heterogeneity: some method choices can affect both estimates and standard errors in the same direction. Our $p$-hacking simulation only partly addresses this mechanism by allowing researchers to change control variables, which can affect both estimates and standard errors. So we model only some of the mechanisms which create problems with inverse-variance weighting and Egger regression in meta-analysis.

The intuitive statistical solution to this problem, often called endogeneity, is to find an instrument for the standard error. A valid instrument is strongly correlated with the standard error, but uncorrelated with the error term in (4)—and thus unrelated to the four sources of endogeneity mentioned above. These are the two assumptions that MAIVE needs in order to relax the crucial exogeneity assumption of meta-analysis models.

The answer beckons. As is evident from Eq. (2), reported variance is a linear function of the inverse of the sample size used in the primary study. The sample size is plausibly robust to selection, or at least it is often harder to collect more data than to $p$-hack the standard error to achieve significance. In observational research, authors typically use as much data as available from the start. The sample size is not estimated, and so it does not suffer from measurement error. The sample size is often not systematically affected by changing methodology and not mechanically related to standardized effect sizes. Indeed, the sample size, unlike the standard error, is often given to the researcher: the very word data means "things given."

We regress the squared reported standard errors on the inverse sample size (Eq. (3)) and plug the fitted values $[\hat{\psi}_0 + \hat{\psi}_1(1/N_i)]$ instead of the variance into the right-hand side of Eq. (4). Thence we obtain the baseline MAIVE estimator. For the baseline MAIVE we choose the instrumented version of PET-PEESE without inverse-variance weights because it works well in simulations and applications. The version with adjusted weights (again, using fitted instead of reported precision) often performs similarly but is more complex. Any meta-analysis technique can be adjusted by the procedure, using $\sqrt{\hat{\psi}_0 + \hat{\psi}_1(1/N_i)}$ instead of reported standard errors. Adjusted standard errors can be retrieved using our `maive` package in R.

## Simulations

Table 2 shows the variants of individual estimators we consider. We always start with the unadjusted variant. Where possible, we consider the adjustment of weights and identification devices separately. So, for PET-PEESE and EK we have 5 different variants: 1) unadjusted, 2) adjusted weights, 3) instrumented SEs, 4) adjusted weights and instrumented SEs, and 5) instrumented SEs and no weights. In variant 3, only the regressor is MAIVE-adjusted, not the weights. In variant 5, weights are dropped altogether for parsimony—note that doing so brings MAIVE closer to a random-effects model that accounts for extreme heterogeneity than to a fixed-effect model. The separation into these five variants is not straightforward for selection models, and we do not pursue it here.

In Fig. 2 we report one set of simulation results: the case of the $p$-hacking scenario with a positive underlying effect size. Here the authors of primary studies run regressions with two variables on the right-hand side and are interested in the slope coefficient on the first variable. A meta-analyst collects the slope coefficients. The vertical axis

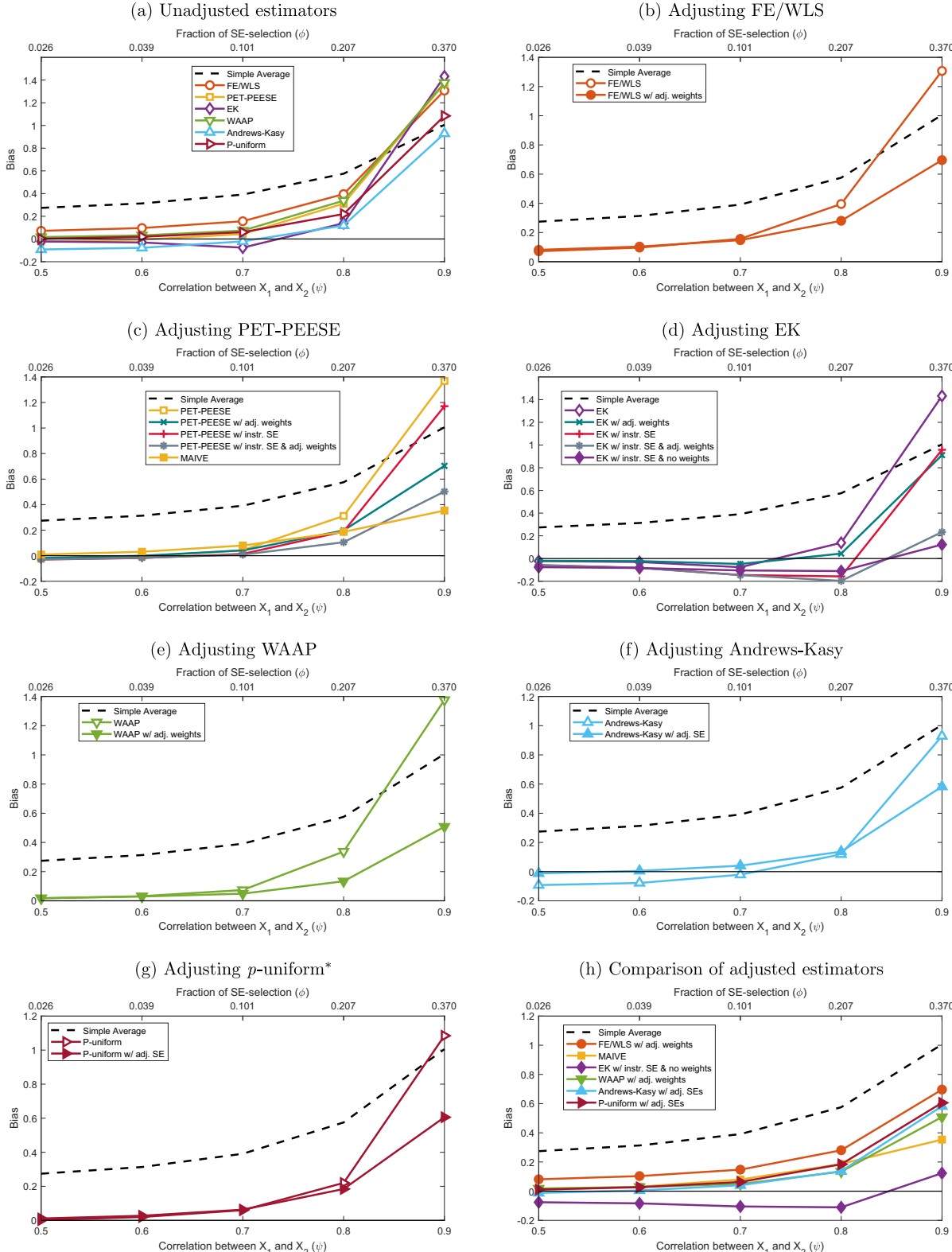

**Fig. 2 | Bias: *p*-hacking selection; MAIVE denotes the estimator's unweighted version.** Notes: The true effect in the simulation is set to 1. The vertical axis shows the bias of meta-analysis estimators. A higher correlation (on the bottom horizontal axis) between the main and control regression variables is associated with more relative importance of spurious precision (top horizontal axis). In the Methods section and the Supplementary Information (SI) we provide details on the simulations. The dashed line shows the performance of the simple mean of published estimates. Panels show (**a**) a comparison of biases for all unadjusted estimators; bias for (**b**) the

fixed effect or weighted least squares estimator with adjustment, (**c**) the adjusted precision-effect test and precision-effect estimate with standard errors, (**d**) the adjusted endogenous kink estimator, (**e**) the adjusted weighted average of adequately powered, (**f**) the adjusted Andrews and Kasy estimator, (**g**) the adjusted p-uniform* method; and (**h**) a comparison of biases for all adjusted estimators. All estimators in panel (**h**) use MAIVE-adjusted standard errors; the default is the MAIVE version of PET-PEESE without weights.

**Table 3 | MAIVE estimates tend to be closer to 0 than PET-PEESE in meta-analyses**

| | All effect sizes | | PET-PEESE significant | |
|---|---|---|---|---|
| | Absolute | % | Absolute | % |
| *(a) All meta-analyses* | | | | |
| \|MAIVE\| > \|PET-PEESE\| | 111 | 35.8 | 63 | 29.9 |
| \|MAIVE\| < \|PET-PEESE\| | 199 | 64.2 | 148 | 70.1 |
| Total | 310 | 100 | 193 | 100 |
| *(b) Meta-analyses with F > 10* | | | | |
| \|MAIVE\| > \|PET-PEESE\| | 87 | 32.6 | 44 | 24.6 |
| \|MAIVE\| < \|PET-PEESE\| | 180 | 67.4 | 135 | 75.4 |
| Total | 267 | 100 | 172 | 100 |
| *(c) Meta-analyses with F > 100* | | | | |
| \|MAIVE\| > \|PET-PEESE\| | 70 | 29.2 | 36 | 22.4 |
| \|MAIVE\| < \|PET-PEESE\| | 169 | 70.7 | 125 | 77.6 |
| Total | 239 | 100 | 151 | 100 |

Notes: The table compares the results of MAIVE and PET-PEESE for the sample of economics meta-analyses[56]. The table separates the cases in which the estimated underlying effect in PET-PEESE is statistically significant at the 5% level (right) and when all PET-PEESE estimates are considered (left). In both cases, MAIVE estimates are typically closer to zero (that is, smaller in absolute value) than PET-PEESE estimates. The difference is larger for statistically significant effects and for meta-analyses with a large *F*-statistic in the first-stage regression of MAIVE; the *F*-statistic therefore measures the strength of the MAIVE instrument. The larger the *F*-statistic, the more reliable MAIVE is, although the difference in performance is not large.

in Fig. 2 measures the bias of meta-analysis estimators relative to the true value of the slope coefficient ($\alpha_1 = 1$). The bottom horizontal axis measures the correlation between the regression variable of interest and a control variable that should be included—but can be replaced by some researchers with a less relevant control, a practice that affects both estimates and their standard errors. The top horizontal axis captures the corresponding importance of selection on standard errors relative to selection on estimates. Technical details are available in the Methods part.

The higher the correlation, the more potential for *p*-hacking via the replacement of the control variable. Replacing the control variable with a less relevant proxy creates an upward omitted-variable bias if, as we assume in the simulation, the signs of $\alpha_2$ and the correlation are the same. A higher correlation causes collinearity and thus increases the potential for selection on standard errors more than proportionally compared to selection on estimates. With a higher correlation and thus more *p*-hacking and also more relative selection on standard errors, the bias of standard meta-analysis estimators increases. Note that even a large correlation still corresponds to a relatively small ratio of selection on standard errors (spurious precision, Taylor's law) relative to selection on estimates (Lombard effect). In the Supplementary Information (Supplementary Table S5) we compute and tabulate this correspondence for different values of the true effect and the correlation.

Eventually, the bias of the unadjusted techniques gets even larger than the bias of the simple unweighted mean. That is, with enough selection on standard errors the simple mean of reported estimates defeats the current estimators that are meant to correct the important bias in the simple mean. Spurious precision can therefore reduce the effectiveness of conventional bias-correction methods, sometimes making them less reliable than the simple mean they are intended to improve upon. MAIVE corrects most of this bias (see panels B-G in Fig. 2 and compare panel A to panel H), and the MAIVE versions with adjusted or omitted weights work similarly well. In this scenario, the MAIVE version of EK works even better than the MAIVE version of PEESE (default MAIVE), but this is not the case in other simulations.

MAIVE performs comparably to conventional estimators if spurious precision is negligible, and dominates unadjusted estimators if spurious precision is important.

In the Supplementary Information (S1, S3–S6) we report the results of more simulation scenarios, both stylized and *p*-hacking, for bias, MSE, and coverage rates—with comparable results in qualitative terms. Even a modest dose of spurious precision can make inverse-variance weighting unreliable and warrants a MAIVE treatment. Note that our simulations allow for substantial heterogeneity among primary studies. By inducing biases in the reported estimates, *p*-hacking causes excess variation in the reported findings (i.e., heterogeneity—$I^2$ of 40–70% depending on simulation context). When additionally allowing for true effect heterogeneity via traditional random effects, $I^2$ surpasses 80% in all contexts. Doing so in the Supplementary Information (S3) does not affect our main results but slightly improves the performance of selection models relative to funnel-based ones.

## Applications

We complement simulations with empirical applications. Our strategy is based on Kvarven et al.[55], who compare meta-analysis results to later preregistered multiple-laboratory replications of the same effect. Kvarven et al. find that common meta-analysis techniques, including those that explicitly correct for publication bias, yield estimates substantially larger than those of replications. The finding has two implications: First, publication bias and *p*-hacking are important on average, create an upward bias in the mean reported results, and should be taken into account by meta-analysts. Second, existing meta-analysis techniques fail to correct enough for these problems. If MAIVE helps reduce meta-analysis estimates and thus bring them closer to replications, it solves a part of the puzzle identified by Kvarven et al. and, at the same time, establishes the empirical relevance of spurious precision.

In the first application we use the dataset of Kvarven et al. focusing on psychology research. However, the dataset does not include sample sizes for individual primary studies, and we need sample sizes for MAIVE correction. Kvarven et al. provide us with additional, previously unpublished data on sample sizes for some meta-analyses. The rest of the required sample sizes we collect manually or obtain by e-mail from the authors of meta-analyses or primary studies. In the end we succeed in collecting sample sizes for all replication–meta-analysis pairs.

Kvarven et al. find that, among meta-analysis estimators, PET-PEESE yields results closest to those of preregistered multiple-laboratory replications. So in our applications, consistently with the previous simulations, we use PET-PEESE as the benchmark and employ its MAIVE version (again, without inverse-variance weights) as the default MAIVE estimator. We find that, when conditions for instrumental variable estimation are met (strong instrument, at least modest sample size), MAIVE further reduces the bias of PET-PEESE in 75% of the cases. More details on estimation and inference are available in the Methods part and in the Supplementary Information (S1).

But the sample of replication–meta-analysis pairs is limited due to the costs of multilab replications. So in the second, more important application we turn to a large dataset of meta-analyses (especially in economics, but also education, finance, business, political science, and sociology) used in Bartos et al.[56]. These meta-analyses typically focus on observational research. As in the previous case, we need sample sizes for individual primary studies (or estimates), and these are often not included in the dataset. We collect as many missing sample sizes as possible manually or obtain them by e-mail from the authors of meta-analyses or primary studies. Technical details are available in the Methods part and in the Supplementary Information (S2).

Our results suggest that, when conditions for instrumental variable estimation are strongly met (239 meta-analyses), MAIVE further reduces the PET-PEESE estimates in more than 70% of the cases. Table 3 shows that the reduction is even stronger if we consider only

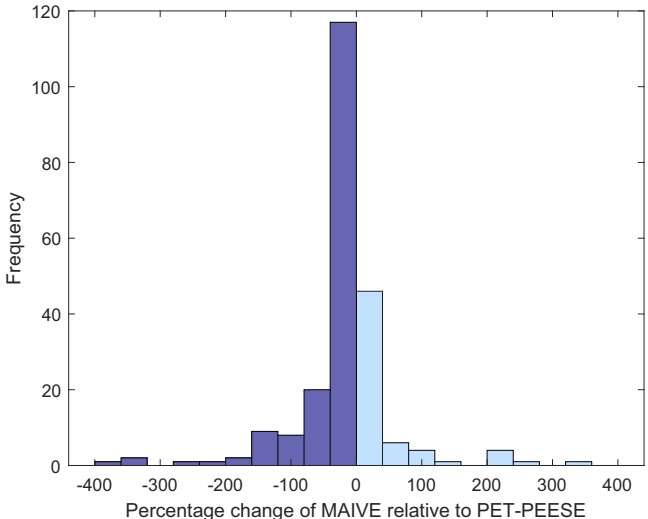

**Fig. 3 | Histogram of the percentage change of MAIVE relative to PET-PEESE.**
Notes: The figure compares the results, in absolute value, of MAIVE and PET-PEESE for the sample of economics meta-analyses[56]. Only meta-analyses with the $F > 100$ from the first-stage regression of MAIVE are included. In most cases, MAIVE adjustment moves PET-PEESE estimates closer to zero.

meta-analyses for which PET-PEESE delivers an estimate statistically significant at the 5% level (two-sided). The reduction also tends to be stronger when the correlation between reported variance and inverse sample size is stronger. Figure 3 shows the main outcome of empirical applications visually: though the pattern is by no means perfect, reductions in PET-PEESE-corrected effects due to MAIVE predominate in meta-analyses of observational research.

## Discussion

Simulations reveal three stylized facts. First, spurious precision can plausibly arise in observational research, for example through *p*-hacking. Second, even a modest degree of spurious precision—introduced in our simulations via selection on standard errors—can create formidable challenges for existing meta-analysis models based on inverse-variance weighting, including those designed to correct for publication bias. Third, the Meta-Analysis Instrumental Variable Estimator (MAIVE) we introduce significantly reduces the resulting bias in meta-analysis.

Empirical applications reveal that spurious precision matters in practice: (i) MAIVE produces results that systematically differ from those of unadjusted meta-analysis models, and (ii) it helps explain the discrepancy identified by Kvarven et al.[55], between meta-analytic estimates and replication results, as MAIVE typically reduces meta-analytic effects. These applications are crucial for corroborating the presence of spurious precision, since no simulation, however comprehensive, can fully capture the complexity of real-world research. Controlled but necessarily stylized simulations are thus complemented by empirical applications that use real data from published studies. Together, they provide converging evidence that spurious precision is an important issue in observational research.

Our results imply that the default use of inverse-variance weights in meta-analysis is problematic. Under classical assumptions, these weights increase the precision of the overall estimate. But when spurious precision is present, inverse-variance weighting can distort inference or even introduce bias. In fact, with enough spurious precision and publication selection, a simple unweighted mean of reported estimates can be less biased than sophisticated inverse-variance-weighted models designed to correct the very bias affecting the mean. In other words, spurious precision can produce more bias than classical publication bias. Our simulations and empirical applications show

that dropping inverse-variance weights often does not harm, and can sometimes improve, the performance of meta-analytic estimators.

The estimator we recommend, MAIVE, builds on PET-PEESE (a variant of Egger regression), but drops inverse-variance weights and replaces the reported variance with the portion explainable by inverse sample size, using an instrumental variables approach. MAIVE has three limitations. First, the instrument should be strong: the correlation between inverse sample size and reported variance needs to be substantial. Our empirical applications show that MAIVE's performance improves with instrument strength. Caution is warranted if the *F*-statistic from the first-stage regression of reported variance on inverse sample size falls below 10. The weak-instrument-robust Anderson-Rubin confidence interval, computed by our `maive` package, should be reported. Second, because MAIVE uses a two-stage estimator, it is relatively imprecise and performs better in larger meta-analyses. Third, MAIVE assumes that sample size is not subject to selection. If meta-analysts suspect that authors systematically choose sample sizes based on expected effect sizes, MAIVE may not offer an advantage over conventional estimators.

When inverse sample size is strongly correlated with reported variance, the number of estimates is at least in the double digits, and sample size is not itself likely to be selected upon, MAIVE can be applied with minimal cost in virtually any meta-analysis. In practice, meta-analysts only need to collect sample sizes from primary studies, a step many already take. Our results show that while MAIVE does not eliminate bias entirely, it outperforms traditional bias-correction estimators when spurious precision is substantial. When spurious precision is negligible, MAIVE performs similarly to standard techniques. It thus serves as a valuable robustness check, even when spurious precision is not initially suspected.

Why not simply replace reported variance with inverse sample size, as proposed in earlier work[37,57–59]? While such a substitution also addresses spurious precision, the instrumental variable approach offers several advantages, as detailed in Supplementary Information (S1). One key advantage is flexibility: the instrumental approach can incorporate other aspects of study design, beyond sample size, that affect precision. Because sample size is rarely a perfect proxy for true variance, MAIVE accounts for this imperfection through Anderson-Rubin confidence intervals. The method can also be extended with additional instruments, such as variables related to clustering, heteroskedasticity adjustment, or identification strategy in primary studies, to improve fit and robustness.

Spurious precision can also be addressed by using the original data from primary studies, as done in individual participant data meta-analysis, the emerging gold standard in medicine[60]. This approach, however, remains infeasible in most observational research due to limited availability of data and code, as well as substantial methodological heterogeneity. In contrast, MAIVE is feasible in most contexts. By dropping inverse-variance weights, MAIVE is conceptually closer to a random-effects meta-analysis model that accounts for extreme heterogeneity than to a fixed-effect model. Given its minimal data requirements and strong performance in the presence of spurious precision, we recommend that MAIVE be routinely reported alongside conventional meta-analysis estimators. The `maive` R package and web interface (spuriousprecision.com) provide accessible tools for implementation.

## Methods
### Stylized selection simulation
This scenario allows us to cleanly separate selection on estimates from selection on standard errors, and the separation forms a useful benchmark and starting point for the more realistic *p*-hacking scenario. All codes and data used in this study are available in GitHub and Zenodo[61].

**Generation of primary data.** The primary data in the stylized scenario are generated according to

$$Y = \alpha_0 + \alpha_1 X + u, \tag{5}$$

where $\alpha_0 = 0$ (without loss of generality), $X \sim U(0, 1)$ and $u \sim N(0, \sigma_u^2)$. The parameter of interest to meta-analysis is $\alpha_1$. Let $i$ index each primary study and let there be $M$ such primary studies, so that $i = 1, 2, ..., M$. Each primary study obtains random samples of size $N_i$ for variables $Y$ and $X$, estimates the regression model specified by Eq. (5), and reports the OLS estimate of $\alpha_1$ and the corresponding standard error.

**Selection.** We simulate a case in which some researchers prefer positive and statistically significant estimates of $\alpha_1$. In the stylized scenario, a fraction $\pi$ of the researchers engage in questionable research practices and are willing to change, by any means necessary, either the reported estimates (E-selection) or the standard errors (SE-selection) in order to inflate the statistical significance of their findings. They do so only when obtaining a positive but statistically insignificant estimate. Hence, when the obtained estimate is either negative or positive but statistically significant, the results are reported without any change. If, on the contrary, the obtained estimate is positive but statistically insignificant, the researcher changes the originally obtained findings with probability $\pi$.

With probability $\phi$, a researcher willing to engage in questionable research practices chooses SE-selection, replacing the obtained standard error by a value that is just enough to achieve statistical significance at the 5% level (two-sided); that is, the reported standard error is $\hat{\alpha}_1/1.96$. With probability $1-\phi$ the researcher chooses instead E-selection, replacing the obtained estimate by a value that is just enough to achieve statistical significance at the 5% level; that is, the reported estimate is $SE(\hat{\alpha}_1) \times 1.96$.

In this environment, therefore, the overall magnitude of publication selection is measured by $\pi$, and the relative importance of SE-selection versus E-selection is measured by $\phi$. Note that, to keep perfect control on $\phi$ as measuring the relative importance of the two types of selection, we assume that researchers do not engage in both types simultaneously. This is also the reason why we assume that negative estimates are not subject to selection; otherwise, a negative estimate would have to become positive, which would necessarily involve E-selection. This restriction will be removed in the *p*-hacking scenario.

**Parameter values and distributions.** We implement this type of selection by means of the parameter values and distributions summarized in Supplementary Table S3 in the Supplementary Information. The number of studies in a meta-analysis is $M = 80$ in line with previous related simulations[7,9]. In Supplementary Information (S4) we also consider simulations with $M = 30$. We assume that primary sample sizes are drawn from a uniform distribution over (30, 1000); in the next section we will calibrate the sample size distribution based on 436 published meta-analyses. We consider three alternative values of $\alpha_1$: zero, one, and two. We interpret these values as representing no effect, a moderate effect, and a large effect, respectively. We assume that the probability of potentially engaging in questionable research practices is $\pi = 0.5$ and let $\phi$ vary from 0 to 1 in steps of 0.25. Note that $\phi = 0$ corresponds to pure E-selection, whereas $\phi = 1$ corresponds to pure SE-selection. Finally, we calibrate $\sigma_u^2 = 3.3$ in order to generate similar effective incidences of selection for $\alpha_1 = 0$ and $\alpha_1 = 2$, which is about 24% in both cases; for $\alpha_1 = 1$ it is a bit larger, at about 32%. (The effective incidence of selection is the overall fraction of findings subject to selection. Note that, in this scenario, effective selection incidence has a hump-shaped profile when graphed against $\alpha_1$. This is because of the assumption of no selection on negative findings. Hence, when $\alpha_1 = 0$, selection incidence is not very high because approximately half of the estimates are negative. It gets higher for $\alpha_1 = 1$, because less estimates are then negative. And it gets lower again for $\alpha_1 = 2$ because more estimates become significantly positive naturally, even without selection.)

**Replications and statistics.** To study the performance of the 7 baseline estimators and their MAIVE variants, we set the number of replications to $R = 2000$. We compute the bias and the mean squared error (MSE) of each estimator by averaging the estimation errors and the squared estimation errors over $R$. Hence, for a generic estimator $z$, the two statistics are given by:

$$\text{Bias}(z) = \frac{1}{R}\sum_{i=1}^{R}(z_i - \alpha_1)$$
$$\text{MSE}(z) = \frac{1}{R}\sum_{i=1}^{R}(z_i - \alpha_1)^2$$

In addition, we also compute the coverage rates of each estimator by counting the number of confidence intervals that contain the true value of $\alpha_1$ as a fraction of the total number of replications. All the results of the simple stylized scenario are discussed in the Supplementary Information (S1).

### *P*-hacking simulation

**Generation of primary data.** In the more realistic *p*-hacking simulation the data generating process for primary studies includes not one but two regressors, $X_1$ and $X_2$:

$$Y = \alpha_0 + \alpha_1 X_1 + \alpha_2 X_2 + u, \tag{6}$$

where, again, $\alpha_0 = 0$ (without loss of generality), $X_1 \sim U(0, 1)$, and $u \sim N(0, \sigma_u^2)$. The second regressor, $X_2$, is a convex combination of $X_1$ and an independent random term $\epsilon \sim N(0, 1)$; i.e., $X_2 = \psi X_1 + (1 - \psi)\epsilon$, where $\psi \in (0, 1)$. Hence, $X_1$ and $X_2$ are positively correlated by construction, this correlation being governed by $\psi$. The parameter of interest to meta-analysis is $\alpha_1$. In some simulations in the Supplementary Information (S3), this parameter is allowed to be random. In this case, $\alpha_1$ refers to its mean and $\sigma_{\alpha_1}^2$ is its variance. The $M$ primary studies each report an OLS estimate and a corresponding standard error of $\alpha_1$ using a sample of size $N_i$. The numerical values of the parameters depend on the selection mechanism assumed and are discussed below.

**Selection.** In this selection scenario, some researchers engage in questionable research practices by manipulating the specification of the model. In particular, we assume that primary studies start by estimating the correctly specified model (Eq. (6)). If the obtained estimate of $\alpha_1$ is not positive and statistically significant in the correctly specified model, then, with probability $\pi$, the dissatisfied authors of such a primary study replace the true control variable $X_2$ by a different control variable, $X_3$. They try many such variables until they find one that 'works,' in the sense of turning the estimate of $\alpha_1$ positive and statistically significant. (We implement this idea by first uniformly drawing a correlation coefficient between $X_2$ and $X_3$, constrained to be positive and less than 0.8. We then generate variable $X_3$ to match this correlation with $X_2$. The maximum correlation of 0.8 is imposed just to save on computing time, since very high correlations do not help the cause of getting statistical significance.)

Replacing $X_2$ by a related but weaker control variable $X_3$ helps achieving statistical significance through both E-selection and SE-selection. E-selection works through the bias it causes on the estimate of $\alpha_1$. Because $X_1$ and $X_2$ are positively correlated, dropping $X_2$ in fact biases the estimate of $\alpha_1$ upwards (omitted-variable bias), making statistical significance more likely. The bias increases with the correlation between $X_1$ and $X_2$ and with the value of $\alpha_2$. The bias is somewhat mitigated by the inclusion of $X_3$. Note that, by inducing biases in the reported estimates, *p*-hacking causes not only publication selection

but also excess variation in the reported findings (i.e., heterogeneity), a feature that characterizes most meta-analyses in observational research. In economics, for example, heterogeneity—rather than the unconditional mean—is often the focus of applied meta-analyses.

The $p$-hacking process also causes SE-selection: replacing $X_2$ by a weaker control decreases collinearity, thus artificially decreasing the SE of the estimate of $\alpha_1$ relative to the SE in the correctly specified model. SE-selection increases with the correlation between $X_1$ and $X_2$, governed by $\psi$, but proportionally more so than E-selection. (To see this, note that E-selection depends on the bias of the estimate of $\alpha_1$. At most, in case $X_2$ is dropped or replaced by an irrelevant control, this bias is given by $\psi\alpha_2$, and thus increases linearly with $\psi$. But SE-selection increases more than linearly with $\psi$. This is because the SE of $\hat{\alpha}_1$ in a model where $X_2$ is included can be written as $c\sqrt{1/(1-\psi^2)}$, where $c$ depends on the variances of $Y$ and $X_1$, on the $R^2$ of the regression, and the sample size, but is invariant to $\psi$. So the SE increases more than linearly with $\psi$, approaching infinity as $\psi$ approaches one.) Thus this scenario still allows us to control the relative magnitude of SE-selection versus E-selection, albeit indirectly and imperfectly. It is not possible to fully decouple the two flavors of selection in this simulation environment, unlike in the stylized scenario.

On a technical note, we limit the number of control variables attempted by $p$-hackers to $H$. If, at the $H$-th attempt, the estimate of $\alpha_1$ remains negative or statistically insignificant, the $p$-hacker gives up the $p$-hacking search and resorts instead to an entirely new dataset, starting the process all over again. We do so because it may be extremely difficult (and time-consuming, from a computational perspective) in some cases to $p$-hack a very negative estimate into a significantly positive one. This is especially the case when $\alpha_1$ is very small, which, by sampling error alone, may generate substantially negative estimates in some datasets.

**Parameter values and distributions.** We implement $p$-hacking selection using the parameter values and distributions summarized in Table S4 in the Supplementary Information. A key parameter in the simulations is $\psi$, the correlation between $X_1$ and $X_2$, since it governs the relative degree of SE-selection versus E-selection. The higher its values, the larger the relative degree of SE-selection. (We can quantify, for each value of $\psi$, the relative importance of SE-selection that would correspond to parameter $\phi$ in the stylized scenario; see below.) We let $\psi$ take on values from 0.5 to 0.9 in steps of 0.1, using the middle value of 0.7 as the baseline value for the calibration of other parameters. In line with related simulation studies[7,9], we assume a meta-analysis of $M = 80$ studies ($M = 30$ in the Supplementary Information S3) and a probability of engaging in publication selection (in this context, the fraction of potential $p$-hackers) of $\pi = 50\%$. The maximum number of $p$-hacking attempts before drawing new data is set at $H = 50$.

Regarding the true effect, we consider the cases where it is nil ($\alpha_1 = 0$) and where it is positive ($\alpha_1 = 1$). We do not separately consider a case of a large effect as in the stylized scenario, because once again the results would be qualitatively comparable to $\alpha_1 = 1$, so we have just one value for the positive effect. We set the remaining parameters (especially $\sigma_u^2$) so that $\alpha_1 = 1$ is neither too small nor too large an effect. If $\sigma_u^2$ is too large, $\alpha_1 = 1$ effectively represents a small effect. Conversely, when $\sigma_u^2$ is too small, $\alpha_1 = 1$ effectively represents a large effect. Moreover, the larger the effective size of $\alpha_1$, the smaller the effective incidence of publication selection, eventually dropping to zero. For $\alpha_1 = 0$, the effective incidence of selection is about 49%. (This assumes that primary studies test the null hypothesis that $\alpha_1 = 0$ using a two-sided test at the 5% level. Hence, the probability of not finding a significantly positive value is 97.5%. Because only half of the studies engage in publication selection, the effective selection incidence is half of this rate, that is, 48.75%.) We then choose $\sigma_u^2$ so that the effective selection incidence for $\alpha_1 = 1$ is half of the incidence for $\alpha_1 = 0$—that is, 24%. (With

$\psi = 0.6$, it is about 21%; with $\psi = 0.8$, it is about 30%). The implied value of $\sigma_u^2$ is 5.06.

Note that we focus on non-standardized effect sizes, regression coefficients. If we focused on a standardized effect size, such as standardized mean difference, we would inevitably introduce a mechanical correlation between estimates and standard errors. This correlation would obscure our simulations because our goal in this paper is to model spurious precision that arises due to selection on standard errors, not other sources of endogeneity of the standard error. The interpretation of the effect size is not important for our simulation as long as the values fall within a range that matters for publication selection. For simplicity, we chose true effect values of 0 a 1 and calibrate the variance of the error term for publication bias to matter for these values. However, in standardized terms, given that the standard deviation of $Y$ is about 2.35 and that of $X_1$ is about 0.29, $\alpha_1 = 1$ implies that one standard deviation increase in $X_1$ increases $Y$ by about $0.29/2.35 = 0.12$ standard deviations. That is, the partial correlation coefficient is about 0.12. While the value may seem low, it is consistent with common meta-analyses in our second empirical application: across hundreds of meta-analyses and 170,900 partial correlations, the mean is 0.16 and the median 0.08. For $\alpha_1 = 2$, the standard deviation of $Y$ increases slightly to about 2.52, so that the standardized effect is $2*0.29/2.52 = 0.23$—i.e., one standard deviation of $X_1$ increases $Y$ by about 0.23 standard deviations—almost twice as large as for $\alpha_1 = 1$.

The $p$-hacking scenario generates heterogeneity. Given $\sigma_u^2$, $\alpha_1$, and $\psi$, the main parameter determining the degree of heterogeneity is $\alpha_2$. Based on the findings of applied meta-analyses, simulation studies[7,9] often assume values of $I^2$ of at least 70%. By setting $\alpha_2 = 2$, we arrive at an $I^2$ of about 73% for $\alpha_1 = 0$ (for $\alpha_1 = 1$, the $I^2$ is about half). When allowing for true effect heterogeneity (Supplementary Information S3), we set $\sigma_{\alpha_1}^2$ at 0.64. This further increases the $I^2$ by about 9 percentage points when $\alpha_1 = 0$; and by about 40 percentage points when $\alpha_1 = 1$. The sample size of a primary study, $N_i$, is drawn from a truncated gamma distribution $\Gamma(a, b)$. Note that the mean of this distribution is given by $ab$ and the variance by $ab^2$. We choose the values of $a$ and $b$ to match the research record. Using a database of 436 meta-analyses in economics and related fields compiled by Chris Doucouliagos[8], we find the medians of the mean and variance of the sample sizes within the individual meta-analyses to be 473 and $588^2$, respectively; using these as target values, we find the required gamma parameters to be $a = 0.65$ and $b = 731$. We truncate the distribution from below, so that a sample size is never smaller than 30.

**Implied relative degree of SE-selection.** As mentioned above, we control the relative degrees of selection on estimates and selection on standard errors indirectly through the parameter $\psi$. In the stylized scenario, this relative degree was controlled directly through $\phi$. Although we cannot control $\phi$ directly here, we can nevertheless infer its size for each value of $\psi$. To do so, start by denoting, for the set of selected estimates, the observed (post-selection, hacked) $t$-statistic of $\hat{\alpha}_1$ by $t = \hat{\alpha}_1/\mathrm{SE}(\hat{\alpha}_1)$ and the original (pre-selection, unhacked) $t$-statistic by $t^* = \hat{\alpha}_1^*/\mathrm{SE}(\hat{\alpha}_1)^*$. Of course, the objective of selection is to increase the size of the $t$-statistic, so $t > t^*$. E-selection implies $\hat{\alpha}_1 > \hat{\alpha}_1^*$ and SE-selection implies $\mathrm{SE}(\hat{\alpha}_1) < \mathrm{SE}(\hat{\alpha}_1)^*$. In the $p$-hacking scenario, however, both types usually occur simultaneously and $\phi$ measures the relative importance of each. Because $t/t^* = (\hat{\alpha}_1/\hat{\alpha}_1^*) \times (\mathrm{SE}(\hat{\alpha}_1)^*/\mathrm{SE}(\hat{\alpha}_1))$, it follows that

$$\ln\left(\frac{t}{t^*}\right) = \ln\left(\frac{\hat{\alpha}_1}{\hat{\alpha}_1^*}\right) + \ln\left(\frac{\mathrm{SE}(\hat{\alpha}_1)^*}{\mathrm{SE}(\hat{\alpha}_1)}\right),$$

which decomposes the amount of publication selection in selected estimates (percent change of the $t$-statistic) into its E-selection

component (given by the first term, the percent increase of $\hat{\alpha}_1$ after selection) and its SE-selection component (given by the second term, the percent decrease in $\text{SE}(\hat{\alpha}_1)$ after selection). Hence, the relative importance of SE-selection can be approximated by the relative size of the second term:

$$\phi = \frac{\ln(\,\text{SE}(\hat{\alpha}_1)^*/\,\text{SE}(\hat{\alpha}_1))}{\ln(t/t^*)}. \qquad (7)$$

On a technical note, we need to impose some restrictions to ensure that $0 \le \phi \le 1$. If, for a particular selected estimate, $\text{SE}(\hat{\alpha}_1) > \text{SE}(\hat{\alpha}_1)^*$, then selection must have occurred entirely through the estimates and we set $\phi = 0$. If, on the other hand, $\hat{\alpha}_1 < \hat{\alpha}_1^*$, then selection must have occurred through the standard errors, and we set $\phi = 1$. Supplementary Information shows the values of $\phi$ corresponding to the various values of $\psi$ (Supplementary Table S5). Clearly, the relative importance of SE-selection increases with $\psi$. All the additional results of the $p$-hacking scenario, on top of those presented in the main text, are discussed in the Supplementary Information (S1).

### Application based on Kvarven et al.
We follow the approach of Kvarven et al.[55], who use three meta-analysis estimators: PET-PEESE (as an example of a widely used funnel-based Egger-type technique), 3PSM (a widely used selection model similar to the Andrews-Kasy model used in our simulations), and Trim & Fill (one of the simplest existing correction techniques). We extend the analysis of Kvarven et al. by including the MAIVE version of each of those three estimators. We are most interested in the comparison of PET-PEESE, which was found by Kvarven et al. to have the smallest bias, with the MAIVE version of PET-PEESE without any weights (the version of MAIVE that we have preferred throughout the manuscript).

We recommend to use the MAIVE adjustment if minimal conditions for normal inference in regression analysis are met and, at the same time, if inverse sample size is a reliably strong instrument for reported variance. Regarding the former, for MAIVE we require each meta-analysis to use at least 30 estimates from primary studies (which, among other things, gives the meta-analyst hope that the regression confidence interval can be reliable). Regarding the latter, to be on the safe side in this application we require that the $F$-statistic from the first-stage regression, regressing reported variance on inverse sample size, is larger than 100. Keane and Neal[27] argue that the commonly used threshold of 10 is often not enough to ensure a strong instrument and valid inference. They recommend researchers use the Anderson-Rubin confidence interval, which is robust to weak instruments and ensures valid inference. In our R package `maive` we allow researchers to obtain this substantially more robust confidence interval so that they can use MAIVE with lower values of the $F$-statistics as well.

All the details on the results are reported in the Supplementary Information (S2). In 8 out of the 15 meta-analyses, both conditions for MAIVE (sample size and instrument strength) are met. Out of these 8 meta-analyses, in 6 cases (75%) is the MAIVE version of PET-PEESE with no weights closer to the replication result than unadjusted PET-PEESE is. The MAIVE versions of Trim & Fill and 3PSM do not behave so well, though: in both cases, the majority of MAIVE estimates are farther from the replication values than the standard estimates are. The $F$ statistics from the first-stage regressions are generally pretty large, being larger than 10 in all cases, larger than 100 in all but 2, and larger than 1000 in more than half (8) of them. The meta-analyses are not always large, however: for instance, 5 of them use fewer than 30 estimates.

The Supplementary Information (S2) also reports the mean absolute deviations of each estimator, with and without the MAIVE adjustment, from the baseline (and arguably unbiased) preregistered multilab replication result. Across all meta-analyses, the MAIVE version of PET-PEESE is, on average, slightly farther from the replication value

than the standard version. This finding holds for both weighted and unweighted MAIVE variants. However, if we confine the comparison to meta-analyses with $F > 100$, with sample sizes satisfying $M > 30$, or both, then the MAIVE versions of PET-PEESE, weighted or unweighted, are closer to the replication values than the standard PET-PEESE version is. Again, in this application the MAIVE adjustment does not improve the performance of Trim & Fill and 3PSM.

### Application based on Bartos et al.
The second MAIVE application is performed on a large dataset compiled by Chris Doucouliagos and used in Bartos et al.[56]. The dataset comprises 613 meta-analyses (especially in economics, but also psychology, education, finance, business, political science, and sociology) and includes 209,766 estimates in total, almost exclusively those from observational studies. This unpublished dataset, which we acquired from the authors, includes estimates and standard errors, also with sample sizes in some cases. Because sample sizes are crucial for the MAIVE adjustment, we attempt to collect them for as many primary studies as possible.

We restrict our analysis to (1) meta-analyses with at least 30 estimates ($M > 30$), (2) estimates based on at least 10 observations, (3) estimates with available sample sizes. Doing so reduces the set of usable meta-analyses to 348. We further censor the data at the 1% and 99% percentiles to limit the influence of extreme outliers. For each of the 348 meta-analyses we apply PET-PEESE and its MAIVE version. In 38 meta-analyses, a negative slope estimate appears in the first-stage regression; these meta-analyses are also discarded, resulting in a final set of 310 meta-analyses.

If SE-selection is present, based on our previous $p$-hacking simulations we expect MAIVE to reduce the estimates of PET-PEESE: MAIVE should be less negative than PET-PEESE when PET-PEESE is negative, and less positive when PET-PEESE is positive. This is what we find and report in the main text (Table 3). MAIVE corrects down PET-PEESE more strongly when the first-stage $F$-statistic is large. When $F > 100$, MAIVE estimates are smaller in absolute value than PET-PEESE estimates in 70.7% of the meta-analyses. It is also important to emphasize how likely it is that sample size is a strong instrument for standard errors: in 267 of the 310 meta-analyses for which a first-stage regression is computed (i.e., 86%), the first-stage $F$ statistic exceeds 10. And in 239 of those (i.e., 77%), the $F$ statistic is larger than 100.

Finally, Fig. 3 in the main body of the paper plots the histogram of the percentage change of MAIVE relative to PET-PEESE. Dark blue bars indicate negative changes, i.e. meta-analyses for which MAIVE is smaller than a positive PET-PEESE estimate or higher than a negative PET-PEESE estimate. The dark bars should dominate if MAIVE is to correct down PET-PEESE estimates. Light blue bars, in contrast, correspond to meta-analyses for which MAIVE exacerbates the PET-PEESE estimate. Clearly, and consistent with the results in Table 3, MAIVE corrects down most PET-PEESE estimates.

### Reporting summary
Further information on research design is available in the Nature Portfolio Reporting Summary linked to this article.

## Data availability
All data used in this study have been deposited in the meta-analysis.cz database at meta-analysis.cz/maive and archived at https://doi.org/10.5281/zenodo.15425605. These are raw data.

## Code availability
All codes used in this study have been deposited in the GitHub database at github.com/meta-analysis-es/maive and archived at https://doi.org/10.5281/zenodo.15425605. The `maive` R package and an interactive web application are available at spuriousprecision.com.

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

## Acknowledgements

We are grateful to Frantisek Bartos, Stephan Bruns, Steven Goodman, John Ioannidis, Stepan Jurajda, Maya Mathur, Shinichi Nakagawa, Bob Reed, Tom Stanley, and Elizabeth Tipton for useful comments that helped us improve the paper. We thank Amanda Kvarven, Josh May, David Rand, and Chris Doucouliagos for sending us additional data on top of their published datasets. Z.I., P.R.D.B., and H.R. acknowledge support from the Czech Science Foundation (grant no. 23-05227M). P.R.D.B. also acknowledges support from the Basque Government Department of Education (grant no. IT1429-22). H.R. also acknowledges support from the Spanish Ministry of Science and Innovation (grant no. PID2020-114646RB-C43). P.R.D.B. and H.R. acknowledge support under grant PID2023-152916NB-I00 financed by MCIN/AEI/10.13039/ 501100011033. T.H. acknowledges support from the Czech Science Foundation (grant no. 24-11583S) and from the Institute for Research on the Socioeconomic Impact of Diseases and Systemic Risks (grant no. LX22NPO5101), funded by the European Union–Next Generation EU.

## Author contributions

Z.I. and T.H. proposed the research idea; P.R.D.B. and H.R. designed the simulations; Z.I. and T.H. collected data for applications; P.R.D.B. and H.R. coded the study and executed the simulations and applications; Z.I. interpreted the results, with assistance from T.H., P.R.D.B., and H.R.; Z.I. and T.H. wrote the main text, with assistance from P.R.D.B. and H.R. Finally, P.R.D.B. and H.R. wrote the Supplementary Information, with assistance from T.H.

## Competing interests

The authors declare no competing interests.
