## [Transparent Peer Review file · Nature Communications]

Spurious Precision in Meta-Analysis of Observational Research

Corresponding Author: Dr Zuzana Irsova

This manuscript has been previously reviewed at another journal. This document only contains information relating to versions considered at Nature Communications. Mentions of the other journal have been redacted.

Version 0:

Reviewer comments:

Reviewer #1

(Remarks to the Author)

I would like to congratulate the authors for the revised version. The manuscript has greatly improved. I was reviewing this article in two rounds for NHB. I read the article again. All my comments are addressed and I do not have additional comments. In my view, this is an article that makes an important point of broad relevance for meta-analyses of observational research. The article provides an intuitive motivation of the issue and demonstrates its relevance using both simulations and empirical applications. The article is readable and accessible for a broad readership.

(Remarks on code availability)

Reviewer #2

(Remarks to the Author)

I previously reviewed this manuscript for [Redacted] (as Referee 2) and was highly supportive of its publication. Upon re-evaluating this third revision for Nature Communications, I remain convinced of its substantial contribution to the literature.

The authors have significantly strengthened their manuscript through extensive revisions and additions, clearly addressing previous comments—including my own, by enhancing clarity, adding comprehensive empirical analyses, and demonstrating the practical value of their proposed MAIVE method. The newly added empirical applications, particularly the comparison with preregistered multilab replications and the large-scale economics meta-analyses, convincingly showcase the method's relevance and practical importance beyond simulation scenarios alone.

The core contribution—identifying and correcting the issue of "spurious precision" in observational research meta-analyses—is important and novel. It addresses a significant gap in meta-analytic methodology that could substantially bias evidence synthesis in observational sciences.

Referee 4 has raised concerns primarily about conceptual and terminological issues surrounding "spurious precision." While their comments are thoughtful and constructive, I find the authors' responses clear and sufficient. Referee 4's concerns about the term "spurious precision" largely stem from distinguishing between precision and bias issues. The authors' revised explanations substantially resolve this ambiguity, especially clarifying how spurious precision interacts with biases in observational meta-analyses. Furthermore, their illustrative scenarios and improved definitions successfully disentangle these complexities, enhancing readability and clarity.

Referee 4's additional comments regarding clustering and the instrumental variable approach have also been adequately addressed. The authors transparently recognize the limitations in addressing clustering (given that the clustering unit often is unclear or unreported), offering practical advice on using sample size as a conservative, widely applicable instrument.

Finally, Referee 4's suggestions to incorporate heterogeneity and clustering simulations are insightful yet demanding. The

authors' extensive empirical validation, particularly their comparison with preregistered replications, reasonably demonstrates MAIVE's robustness even under realistic conditions of substantial heterogeneity.

In summary, this manuscript significantly advances the methodology of meta-analysis for observational research. The revisions clearly address critical comments, and while Referee 4's suggestions are thoughtful, some requests for further simulations may extend beyond practical necessity, especially given the extensive empirical validation already provided (such simulation would be another paper).

I strongly recommend publication. This paper offers valuable insights and practical solutions to an important methodological issue in observational research synthesis.

(Remarks on code availability)

I have reviewed it earlier and it works

Reviewer #5

(Remarks to the Author)

I want to note that I was invited to review this manuscript after three rounds of revision, following extensive feedback from four referees. Given the depth and breadth of prior comments and the authors' thorough responses, offering additional input while respecting the substantial work already done was not straightforward.

The topic addressed by the authors is very important and sometimes overlooked, especially given the major focus on p-hacking related to the parameter value compared to the precision. That said, I found the paper extremely interesting and methodologically valuable. The authors have done an excellent job presenting a complex method and simulation framework in an intuitive and accessible way. The clarity of exposition, particularly in how the rationale behind MAIVE is communicated, deserves recognition.

From the revision history and current version, it is clear that the authors have taken the reviewers' suggestions seriously. The theoretical framework, simulation setup, empirical applications, and even the software implementation have been completely addressed. Based on this, and considering my expertise, I do not have any major concerns or further suggestions.

I do, however, offer a few minor comments that the authors may consider to improve clarity and accessibility further:

- 1. Clarifying "selection" vs. "publication Bias"** . While the manuscript provides a careful definition of selection on standard errors and estimates, the discussion around funnel plot asymmetry could be further clarified. In particular, it may help to explicitly distinguish between asymmetry arising from omitted studies (i.e., traditional publication bias) and asymmetry caused by manipulated or selectively reported statistics, which is the focus of this paper.
- 2. Applicability to standardized effect sizes** . The choice to simulate unstandardized effect sizes is well explained—mainly to remove mechanical correlations with standard errors. Still, a brief statement reassuring readers that the MAIVE method can also be applied in settings involving standardized effect sizes might help broaden its perceived applicability.
- 3. Emphasizing the sample size as core information**. It may be helpful to underscore that MAIVE primarily relies on total sample size as an instrument, regardless of the model specification used in primary studies (e.g., multilevel or clustered designs). This point has been made but could be reinforced for clarity.
- 4. Consider adding a schematic overview of the Simulation Setup**. The simulation design is both elegant and very useful for future simulation studies. Including a schematic or diagram summarizing the simulation process would enhance understanding and highlight this contribution, which is valuable also beyond the context of MAIVE.
- 5. Clarify labels in Figure 2 and supplementary figures** . As these figures are central to the paper, a slightly more detailed caption would help guide the reader. For instance, does "MAIVE" refer to the default unweighted version? In panel (h), are all methods MAIVE-adjusted? And does "w/ adj. SE" indicate the use of MAIVE adjustment within each estimator? Clarifying these points would make the results easier to interpret for a broader audience.
- 6. Code clarity** . I have found the organization of the online website and R package a little bit confusing. The website link for the R package is not linking the Github package but a zip file. A suggestion for the future setup could be to develop everything as an R package with extra documentation and examples as vignettes within the package. In addition also considering proposing the implementation into the `metafor` package or implementing a clear communication between the `rma` class and the MAIVE methods could be very useful. Similar to what the `robust` function is doing in `metafor` for the sandwich variance estimator methods. Something like a general `maive()` function taking in input an `rma`/`rma.mv` fitted object and re-fitting and or adjusting accordingly.

(Remarks on code availability)

I have reviewed the R code provided into the "R code for MAIVE" folder. I have some comments for the future development of the package, not for the paper.

The maive.r file has some small problems:

```
if (!require('rstudioapi')) install.packages('rstudioapi'); library('rstudioapi')
if (!require('readxl')) install.packages('readxl'); library('readxl')
setwd(dirname(getActiveDocumentContext())$path)
dat <- read_excel("inputdata.xlsx")
```

This part is very R Studio-specific, especially the `setwd(dirname(getActiveDocumentContext())$path)` part. I highly suggest to use R Projects or relying assuming that the working directory is automatically adjusted (from the used IDE or manually from the user)

There are some typos `source("maivefunction.R")` need the lowercase `.r` (as the filename)

The github package is not really an R-style package. I highly suggest to implement some printing/summary methods to show the MAIVE results and organize everything (also example data) as a proper R package (<https://r-pkgs.org/>). This can dramatically improve the usability of the functions.

In particular, avoid including `require()` statements within the functions but, if within an R package framework, relying on the DESCRIPTION file with explicit dependencies that will be installed with the package itself.

RESPONSE TO REVIEWERS' COMMENTS

NCOMMS-25-11822-T

"Spurious Precision in Meta-Analysis of Observational Research"

by Irsova, Bom, Havranek, Rachinger

Our responses to referees' comments are typeset in bold. Note that Referee 1 and Referee 2 do not raise any comments in this round.

Reviewer #1 (Remarks to the Author):

I would like to congratulate the authors for the revised version. The manuscript has greatly improved. I was reviewing this article in two rounds for NHB. I read the article again. All my comments are addressed and I do not have additional comments. In my view, this is an article that makes an important point of broad relevance for meta-analyses of observational research. The article provides an intuitive motivation of the issue and demonstrates its relevance using both simulations and empirical applications. The article is readable and accessible for a broad readership.

Authors' response: Thank you for your kind words and useful comments in previous rounds!

Reviewer #2 (Remarks to the Author):

I previously reviewed this manuscript for **[Redacted]** (as Referee 2) and was highly supportive of its publication. Upon re-evaluating this third revision for Nature Communications, I remain convinced of its substantial contribution to the literature.

The authors have significantly strengthened their manuscript through extensive revisions and additions, clearly addressing previous comments—including my own, by enhancing clarity, adding comprehensive empirical analyses, and demonstrating the practical value of their proposed MAIVE method. The newly added empirical applications, particularly the comparison with preregistered multilab replications and the large-scale economics meta-analyses, convincingly showcase the method's relevance and practical importance beyond simulation scenarios alone.

The core contribution—identifying and correcting the issue of "spurious precision" in observational research meta-analyses—is important and novel. It addresses a significant gap in meta-analytic methodology that could substantially bias evidence synthesis in observational sciences.

Referee 4 has raised concerns primarily about conceptual and terminological issues surrounding "spurious precision." While their comments are thoughtful and constructive, I find the authors' responses clear and sufficient. Referee 4's concerns about the term "spurious precision" largely stem from distinguishing between precision and bias issues. The authors' revised explanations substantially resolve this ambiguity, especially clarifying how spurious precision interacts with biases in observational meta-analyses. Furthermore, their illustrative scenarios and improved definitions successfully disentangle these complexities, enhancing readability and clarity.

Referee 4's additional comments regarding clustering and the instrumental variable approach have also been adequately addressed. The authors transparently recognize the limitations in addressing clustering (given that the clustering unit often is unclear or unreported), offering practical advice on using sample size as a conservative, widely applicable instrument.

Finally, Referee 4's suggestions to incorporate heterogeneity and clustering simulations are insightful yet demanding. The authors' extensive empirical validation, particularly their comparison with preregistered replications, reasonably demonstrates MAIVE's robustness even under realistic conditions of substantial heterogeneity.

In summary, this manuscript significantly advances the methodology of meta-analysis for observational research. The revisions clearly address critical comments, and while Referee 4's suggestions are thoughtful, some requests for further simulations may extend beyond practical necessity, especially given the extensive empirical validation already provided (such simulation would be another paper).

I strongly recommend publication. This paper offers valuable insights and practical solutions to an important methodological issue in observational research synthesis.

Reviewer #2 (Remarks on code availability):

I have reviewed it earlier and it works

Authors' response: We are grateful to the referee for the praise and the time they devoted to reading our paper in this and previous rounds of review.

Reviewer #5 (Remarks to the Author):

I want to note that I was invited to review this manuscript after three rounds of revision, following extensive feedback from four referees. Given the depth and breadth of prior comments and the authors' thorough responses, offering additional input while respecting the substantial work already done was not straightforward.

The topic addressed by the authors is very important and sometimes overlooked, especially given the major focus on p-hacking related to the parameter value compared to the precision. That said, I found the paper extremely interesting and methodologically valuable. The authors have done an excellent job presenting a complex method and simulation framework in an intuitive and accessible way. The clarity of exposition, particularly in how the rationale behind MAIVE is communicated, deserves recognition.

From the revision history and current version, it is clear that the authors have taken the reviewers' suggestions seriously. The theoretical framework, simulation setup, empirical applications, and even the software implementation have been completely addressed. Based on this, and considering my expertise, I do not have any major concerns or further suggestions.

Authors' response: Thank you for your kind words and your time!

I do, however, offer a few minor comments that the authors may consider to improve clarity and accessibility further:

1. Clarifying "selection" vs. "publication Bias". While the manuscript provides a careful definition of selection on standard errors and estimates, the discussion around funnel plot asymmetry could be further clarified. In particular, it may help to explicitly distinguish between asymmetry arising from omitted studies (i.e., traditional publication bias) and asymmetry caused by manipulated or selectively reported statistics, which is the focus of this paper.

Authors' response: We now explicitly distinguish in the main text between p-hacking and publication bias: "The separation is less clean in the more realistic p-hacking scenario but can be mapped back to the stylized scenario. Note that both differ from the standard scenario of publication bias, where different studies have different probabilities of publication but are not manipulated or p-hacked." (p. 3)

2. Applicability to standardized effect sizes. The choice to simulate unstandardized effect sizes is well explained—mainly to remove mechanical correlations with standard errors. Still, a brief statement reassuring readers that the MAIVE method can also be applied in settings involving standardized effect sizes might help broaden its perceived applicability.

Authors' response: This is also a good point. We now write in the main text: "Note that while we work with comparable regression coefficients to avoid the mechanical correlation between estimates and standard errors that arises with standardized effect sizes, MAIVE can be applied in settings involving standardized effect sizes. In fact, this mechanical correlation is another reason why MAIVE is particularly useful in such cases." (p. 4)

3. Emphasizing the sample size as core information. It may be helpful to underscore that MAIVE primarily relies on total sample size as an instrument, regardless of the model specification used in primary studies (e.g., multilevel or clustered designs). This point has been made but could be reinforced for clarity.

Authors' response: We now write in the main text: "MAIVE replaces, in all meta-analytic contexts, the reported variance with the portion that can be explained by the inverse of the total sample size used in the primary study, regardless of the model specification in the original studies (e.g., whether they use multilevel or clustered designs)." (p. 6)

4. Consider adding a schematic overview of the Simulation Setup. The simulation design is both elegant and very useful for future simulation studies. Including a schematic or diagram summarizing the simulation process would enhance understanding and highlight this contribution, which is valuable also beyond the context of MAIVE.

Authors' response: We have created a schematic overview and included it in the Supplementary Information (Figure S5). Thank you for the suggestion!

5. Clarify labels in Figure 2 and supplementary figures. As these figures are central to the paper, a slightly more detailed caption would help guide the reader. For instance, does "MAIVE" refer to the default unweighted version? In panel (h), are all methods MAIVE-adjusted? And does "w/ adj. SE" indicate the use of MAIVE adjustment within each

estimator? Clarifying these points would make the results easier to interpret for a broader audience.

Authors' response: We change the label of Figure 2 to include the definition of the MAIVE estimator used in the figure (default unweighted version). In the notes to the table we explain the remaining issues mentioned by the referee.

6. Code clarity. I have found the organization of the online website and R package a little bit confusing. The website link for the R package is not linking the Github package but a zip file. A suggestion for the future setup could be to develop everything as an R package with extra documentation and examples as vignettes within the package. In addition also considering proposing the implementation into the `metafor` package or implementing a clear communication between the `rma` class and the MAIVE methods could be very useful. Similar to what the `robust` function is doing in `metafor` for the sandwich variance estimator methods. Something like a general `maive()` function taking in input an `rma`/`rma.mv` fitted object and re-fitting and or adjusting accordingly.

Reviewer #5 (Remarks on code availability):

I have reviewed the R code provided into the "R code for MAIVE" folder. I have some comments for the future development of the package, not for the paper.

The maive.r file has some small problems:

```
if (!require('rstudioapi')) install.packages('rstudioapi'); library('rstudioapi')
if (!require('readxl')) install.packages('readxl'); library('readxl')
setwd(dirname(getActiveDocumentContext()$path))
dat <- read_excel("inputdata.xlsx")
```

This part is very R Studio-specific, especially the `setwd(dirname(getActiveDocumentContext()$path))` part. I highly suggest to use R Projects or relying assuming that the working directory is automatically adjusted (from the used IDE or manually from the user)

There are some typos `source("maivefunction.R")` need the lowercase `.r` (as the filename)

The github package is not really an R-style package. I highly suggest to implement some printing/summary methods to show the MAIVE results and organize everything (also example data) as a proper R package (<https://r-pkgs.org/>). This can dramatically improve the usability of the functions.

In particular, avoid including `require()` statements within the functions but, if within an R package framework, relying on the DESCRIPTION file with explicit dependencies that will be installed with the package itself.

Authors' response: Thank you for your very useful comments on how our future R package can be designed! We will do as you suggest (and will also create an app that will make it easy to apply MAIVE). The work on the package and app, separate from our work on this paper, will take us some time.